# Enforcing Hard Constraints with Soft Barriers: Safe Reinforcement Learning in Unknown Stochastic Environments

## Abstract

It is quite challenging to ensure the safety of reinforcement learning (RL) agents in an unknown and stochastic environment under hard constraints that require the system state not to reach certain specified unsafe regions. Many popular safe RL methods such as those based on the Constrained Markov Decision Process (CMDP) paradigm formulate safety violations in a cost function and try to constrain the expectation of cumulative cost under a threshold. However, it is often difficult to effectively capture and enforce hard reachability-based safety constraints indirectly with such constraints on safety violation cost. In this work, we leverage the notion of barrier function to explicitly encode the hard safety constraints, and given that the environment is unknown, relax them to our design of *generative-model-based soft barrier functions*. Based on such soft barriers, we propose a safe RL approach that can jointly learn the environment and optimize the control policy, while effectively avoiding the unsafe regions with safety probability optimization. Experiments on a set of examples demonstrate that our approach can effectively enforce hard safety constraints and significantly outperform CMDP-based baseline methods in system safe rate measured via simulations.

## 1 Introduction

Reinforcement learning (RL) (Sutton & Barto, 2018) has shown promising successes in learning complex policies for games (Silver et al., 2018), robots (Zhao et al., 2020), and recommender systems (Afsar et al., 2021), by maximizing a cumulative reward objective as the optimization goal. However, real-world safety-critical applications, such as autonomous cars and unmanned aerial vehicles (UAVs), still hesitate to adopt RL policies due to safety concerns. In particular, these applications often have hard safety constraints that require the system state not reach certain specified unsafe regions, e.g., autonomous cars not deviating into adjacent lanes or UAVs not colliding with trees. And it is very challenging to learn a policy via RL that can meet such hard safety constraints, especially when the environment is stochastic and unknown.

In the literature, the Constrained Markov Decision Process (CMDP) (Altman, 1999) is a popular paradigm for addressing RL safety. Common CMDP-based methods encode safety constraints through a cost function of safety violations, and reduce the policy search space to where the expectation of cumulative discounted cost is less than a threshold. And various RL algorithms are proposed to adaptively solve CMDP through the primal-dual approach for the Lagrangian problem of CMDP. However, it is often hard for CMDP-based methods to enforce reachability-based hard safety constraints (i.e., system state not reaching unsafe regions) by setting *indirect* constraints on the expectation of cumulative cost. In particular, while reachability-based safety constraints are defined on the system state at each time point (i.e., each point on the trajectory), the CMDP constraints only enforce the cumulative behavior. In other words, the cost penalty on the system visiting the unsafe regions at certain time point may be offset by the low cost at other times. There is a recent CMDP approach addressing hard safety constraints by using the indicator function for encoding failure probability (Wagener et al., 2021), but it requires a safe back-up policy for intervention, which is difficult to achieve in unknown environments. Safe exploration with hard safety constraint has been studied in (Wachi et al., 2018; Turchetta et al., 2016; Moldovan & Abbeel, 2012). However,

they focus on discrete state and action spaces where the hard safety constraints are defined as a set of unsafe state-action pairs that cannot be visited, different from our continuous control setting.

On the other hand, current control-theoretical approaches for model-based safe RL often try to leverage formal methods to handle hard safety constraints, e.g., by establishing safety guarantees through barrier functions or control barrier functions (Luo & Ma, 2021), or by shielding mechanisms based on reachability analysis to check whether the system may enter the unsafe regions within a time horizon (Bastani et al., 2021). However, these approaches either require explicit known system models for barrier or shielding construction, or an initial safe policy to generate safe trajectory data in a deterministic environment. They cannot be applied to our unknown stochastic environments.

To overcome the above challenges, we propose a safe RL framework by encoding the hard safety constraints via the learning of a *generative-model-based soft barrier function*. Specifically, we formulate and solve a novel bi-level optimization problem to learn the policy with joint soft barrier function learning, generative modeling, and reward optimization. The soft barrier function provides a guidance for avoiding unsafe regions based on safety probability analysis and optimization. The generative model accesses the trajectory data from the environment-policy closed-loop system with stochastic differential equation (SDE) representation to learn the dynamics and stochasticity of the environment. And we further optimize the policy by maximizing the total discounted reward of the sampled synthetic trajectories from

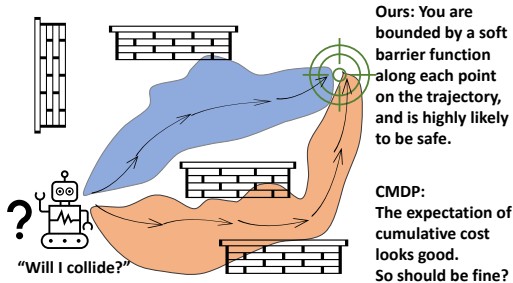

Figure 1: An RL-based robot navigation example that shows the difference between our approach and CMDP-based ones in encoding the hard safety constraints.

the generative model. This joint training framework is fully differentiable and can be efficiently solved via the gradients. Compared to CMDP-based methods, our approach more directly encodes the hard safety constraints along each point of the agent trajectory through the soft barrier function, as shown in Figure 1. While given the unknown stochastic environment, our approach cannot provide a hard barrier and hence no deterministic safety guarantee, experimental results demonstrate that in simulations, ours can significantly outperform the CMDP-based baselines in system safe rate.

The paper is organized as follows. Section 2 introduces related works, Section 3 presents our approach, including the bi-level optimization formulation, our safe RL algorithm with generative modeling, soft barrier function learning, and policy optimization to solve the formulation, and theoretical analysis of safety probability. Section 4 shows the experiments and Section 5 concludes the paper.

## 2 RELATED WORK

**Safe RL by CMDP:** CMDP-based methods encode the safety violation as a cost function and set constraints on the expectation of cumulative discounted total cost. The primal-dual approaches have been widely adopted to solve the Lagrangian problem of constrained policy optimization, such as PDO (Chow et al., 2017), OPDOP (Ding et al., 2021), CPPO (Stooke et al., 2020), FOCOPS (Zhang et al., 2020), and CRPO (Xu et al., 2021). Other works leverage a world model learning (As et al., 2021) or the Lyapunov function to solve the CMDP (Chow et al., 2018), or add a safety layer for the safety constraint (Dalal et al., 2018). However, the constraints in CMDP cannot directly encode the hard reachability-based safety properties, which hinders its application to many safety-critical systems. A recent CMDP-based work uses the indicator function for encoding failure probability as hard safety constraints, but it requires a safe backup policy for intervention (Wagener et al., 2021).

**Model-based Safe RL by Formal Methods:** Formal analysis and verification techniques have been proposed in model-based safe RL to enforce the system not reach unsafe regions. Some works develop shielding mechanisms with a backup policy based on reachability analysis (Shao et al., 2021; Li & Bastani, 2020; Bastani et al., 2021). Other works adopt (control) barrier functions or (control) Lyapunov functions for provable safety (Emam et al., 2021; Choi et al., 2020; Cheng et al., 2019; Wang et al., 2022; Ma et al., 2021; Luo & Ma, 2021; Berkenkamp et al., 2017; Taylor et al.,

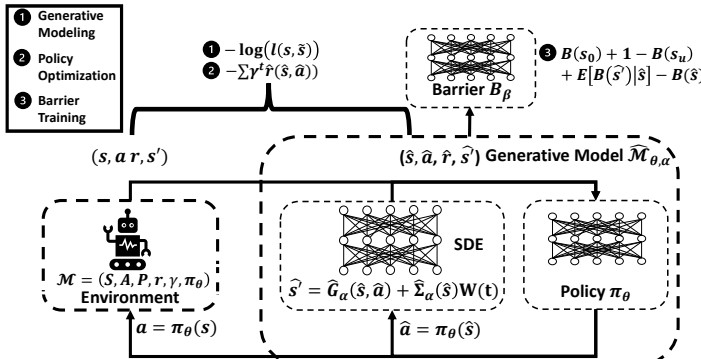

Figure 2: The overview of our safe RL framework based on a generative-model-based soft barrier function. The real environment and generative model share the learning policy and the generative model is abstracted as a discrete-time stochastic differential equation.

2020). Moreover, recent work (Yu et al., 2022) adopts reachability analysis with CMDP to compute safe feasible sets. However, these methods either require known dynamics, a safe initial/backup policy, or human intervention, and thus do not apply to our setting.

**Barrier Function for Safety:** Barrier function is introduced as a safety certificate afflicted to the control policy for deterministic and stochastic systems (Prajna & Jadbabaie, 2004; Prajna et al., 2004). In classical control, finding a barrier function is time-consuming and requires a lot of manual effort, where a common idea is to relax the conditions of barrier function into optimization formulations such as linear programming (Yang et al., 2016), quadratic programming (Ames et al., 2016), and sum-of-square programming (Wang et al., 2022). However, these optimization-based approaches can hardly scale to high-dimensional systems. To this end, recent works have shown great promise in jointly training barrier function and safe policy by neural network representation for better scalability (Qin et al., 2021). Our approach leverages the paradigm of barrier function, but develops the concept of soft barrier to address unknown stochastic environments.

**RL with Generative Model:** Previous works of generative-model-based RL mainly focus on sample efficiency and policy optimization for the total expected return (Agarwal et al., 2020b; Li et al., 2020a; Tirinzoni et al., 2020). Some works (HasanzadeZonuzy et al., 2021; Maeda et al., 2021) address safe RL by CMDP with a generative model, but only solve the tabular discrete state and action space. Besides policy optimization, the generative model in our framework also plays an important role in building a soft barrier function to facilitate the probabilistic safety analysis and optimization.

## 3 OUR APPROACH

In this section, we present our framework for safe RL in unknown stochastic environment that enforces hard safety constraints with soft barrier functions. In Section 3.1, we present our bi-level optimization formulation for the problem, which maximizes a total expected return while trying to avoid unsafe regions. Specifically, we encode the hard safety constraints with a novel generative-model-based soft barrier function in the lower problem and maximize the performance with generative model learning in the upper problem. We then present our safe RL algorithm to solve the bi-level optimization formulation, by jointly learning the generative model (Section 3.2), soft barrier function (Section 3.3), and policy optimization (Section 3.4) via first-order gradient, as shown in Figure 2. We conduct theoretical analysis for safety probability of the learned policy in Section 3.5.

### 3.1 BILEVEL OPTIMIZATION PROBLEM FORMULATION FOR SAFE RL

We assume that the environment can be abstracted as a finite-horizon continuous MDP $\mathcal{M} \sim (\mathcal{S}, \mathcal{A}, \mathcal{P}, r, \gamma, \pi)$, where $\mathcal{S} \in \mathbb{R}^n$ represents the continuous state space, $\mathcal{A} \in \mathbb{R}^m$ indicates the continuous action space, and the function class $\mathcal{P} : \mathcal{S} \times \mathcal{A} \times \mathcal{S} \to [0, 1]$ denotes the unknown continuous and smooth stochastic environment dynamics without jump condition. The rewards function $r(s, a) : \mathcal{S} \times \mathcal{A} \to \mathbb{R}$ is known and the discount factor $\gamma \in [0, 1]$. A deterministic continuous NN-based policy $\pi_\theta : \mathcal{S} \to \mathcal{A}$ maps the states $s(t) \in \mathcal{S}$ to an action $a(t) \in \mathcal{A}$ at

---

**Algorithm 1** Safe RL with the Generative-model-based Soft Barrier Function

---

**Input**: Unknown environment $\mathcal{M}$, initial policy $\pi_\theta$, generative model $\hat{\mathcal{M}}_{\theta,\alpha}$, barrier network $B_\beta$.
**Parameter**: $[\theta, \alpha, \beta]$.
**Output**: Policy $\pi_\theta$ with $\hat{\mathcal{M}}_{\theta,\alpha}$ based soft barrier function $B_\beta$.
 1: For $k$ in $0, \cdots, N$
 2:   For $i$ in $0, \cdots, M$
 3:     Sample processes $\tau_\theta^i$ by policy $\pi_\theta$ with $\mathcal{M}$ and synthetic processes $\hat{\tau}_{\theta,\alpha}^i$ by $\pi_\theta$ with $\hat{\mathcal{M}}_{\theta,\alpha}$.
 4:     Compute generative loss function $\mathcal{L}_g$ with $\tau_\theta^i$ and $\hat{\tau}_{\theta,\alpha}^i$ as in Equation 3, $\alpha \leftarrow \alpha - \frac{\partial \mathcal{L}_g}{\partial \alpha}$.
 5:   Compute barrier function loss $\mathcal{L}_B$ by sampling synthetic $\hat{\tau}_{\theta,\alpha}^k$ as in Equation 4.
 6:   Compute total discount reward $\hat{J}(\pi_\theta)$ by sampling synthetic $\hat{\tau}_{\theta,\alpha}^k$ as in Equation 5.
 7:   $\theta \leftarrow \theta - \frac{\partial \mathcal{L}_B}{\partial \theta} + \frac{\partial \hat{J}}{\partial \theta}$ , $\beta \leftarrow \beta - \frac{\partial \mathcal{L}_B}{\partial \beta}$.

---

time $t$ as $a(t) = \pi_\theta(s(t))$, where $s(t)$ is a random variable at timestep $t$. The environment has several known spaces, i.e., the state space $S \subset \mathcal{S}$, the initial space $S_0 \subset \mathcal{S}$, and the unsafe space $S_u \subset \mathcal{S}$. The RL objective is to maximize the total discounted expected return as $\max_{\pi_\theta} J := \mathbb{E}_{s(0) \in S_0, P(s'|s,a)} \left[ \sum_{t=0}^T \gamma^t r(s(t), a(t)) \right]$ with $P \in \mathcal{P}$. To encode the hard safety constraint $S_u$, we formulate a bi-level optimization problem for our framework as the following. We use ^ to denote the elements related to the generative model.

**Definition 1** *(Bi-level Optimization Problem for Safe RL)*

$$\max_{\theta,\alpha} J(\pi_\theta) - \lambda \eta^*(\theta, \alpha)^2 - \mathcal{L}_g(\tau_\theta, \hat{\tau}_{\theta,\alpha}),$$

*where $\eta^*(\theta, \alpha)$ is the optimal objective to a lower-level problem of the generative-model-based soft barrier function with $\hat{s}$ as the state in the generative model:*

$$\min_\beta \eta \ \text{ s.t.} \begin{cases} B_\beta(\hat{s}) \geq 0, \forall \, \hat{s} \in S, \ \ B_\beta(\hat{s}) \geq 1, \forall \, \hat{s} \in S_u, \ \ B_\beta(\hat{s}) \leq \eta, \forall \, \hat{s} \in S_0, \\ \mathbb{E}\left[B_\beta(\hat{s}(t+1)) | \hat{s}(t)\right] \leq B_\beta(\hat{s}(t)), \hat{s}(t+1) = \hat{\mathcal{M}}_{\theta,\alpha}(\hat{s}(t)), \forall t \in [0,T] \forall \hat{s} \in S \setminus S_u, \end{cases} \tag{1}$$

where $\theta$ is the parameter of policy $\pi$. $\alpha$ is the parameter of the generative model $\hat{\mathcal{M}}_{\theta,\alpha} = (\hat{G}_\alpha, \hat{\Sigma}_\alpha)$, which is a stochastic differential equation (SDE) with $\hat{G}_\alpha$ as the drift function and $\hat{\Sigma}_\alpha$ as the diffusion function for the stochasticity, as shown later in Equation 2. $\lambda \geq 0$ is a penalty multiplier. $\tau_\theta := \{s(0), s(1), \cdots, s(T)\}$ and $\hat{\tau}_{\theta,\alpha} := \{\hat{s}(0), \hat{s}(1), \cdots, \hat{s}(T)\}$ are the sampled realizations of stochastic processes (trajectories) from the environment and from the generative model by the policy $\pi_\theta$, respectively. $\beta$ is the parameter of the generative-model-based soft barrier function $B_\beta : \mathbb{R}^n \to \mathbb{R}^+$. We encode the hard safety constraint by the generative-model-based soft barrier function $B_\beta$ in the lower problem, which minimizes $\eta^*(\theta, \alpha)$ as the upper bound of the unsafe probability for $\hat{\mathcal{M}}_{\theta,\alpha}$ in Section 3.3. The upper problem aims to optimize the policy's expected return $J(\pi_\theta)$ and learn the generative model by the maximum likelihood loss $\mathcal{L}_g(\tau_\theta, \hat{\tau}_{\theta,\alpha})$ between the processes $\tau_\theta$ and $\hat{\tau}_{\theta,\alpha}$ as shown later in Equation 3. Moreover, the upper problem penalizes $\eta^*(\theta, \alpha)$, which can back propagate the gradient information through $\hat{\mathcal{M}}_{\theta,\alpha}$ to $\pi_\theta$ for pushing the agent to avoid $S_u$ in the environment MDP $\mathcal{M}$ as much as possible if $\hat{\mathcal{M}}_{\theta,\alpha}$ behaves similar to $\mathcal{M}$.

We can compute the gradient from $\eta^*(\theta, \alpha)$ for $\pi_\theta$ through $\hat{\mathcal{M}}_{\theta,\alpha}$ with current auto-differential tools. This cannot be done in $\mathcal{M}$ as it is unknown. Therefore, the overall bi-level problem is end-to-end differentiable and can be solved efficiently. Figure 2 shows how the components in our framework interact with each other. The overall algorithm to solve the bi-level problem is shown in Algorithm 1. Next, we are going to introduce the details of each module.

### 3.2 GENERATIVE MODELING

The role of the generative model in our framework is two folds: (1) Because the barrier function requires an environment model to encode the hard safety constraints, the generative model serves

as a surrogate model to build this barrier function, where $\eta^*(\theta, \alpha)$ propagates the gradient to $\pi_\theta$ through $\hat{\mathcal{M}}_{\theta,\alpha}$ for improving system safety. (2) The generative model can generate synthetic process (trajectory) $\hat{\tau}_{\theta,\alpha}$ to optimize the performance of the policy efficiently by gradient propagation.

We learn the generative model $\hat{\mathcal{M}}_{\theta,\alpha}$ as a discrete-time SDE to capture the dynamics and stochasticity of the environment and serve as a base for the construction of the soft barrier function:

$$\hat{\mathcal{M}}_{\theta,\alpha} : \hat{s}(t+1) = \hat{G}_\alpha(\hat{s}(t), \pi_\theta(\hat{s}(t))) + \hat{\Sigma}_\alpha(\hat{s}(t))W(t), \tag{2}$$

where $\hat{G}_\alpha : \mathbb{R}^n \times \mathbb{R}^m \to \mathbb{R}^n$ is an unknown drift function, $\hat{\Sigma}_\alpha : \mathbb{R}^n \to \mathbb{R}^{n \times d}$ is an unknown $n \times d$ matrix based on $\hat{s}$, and $W(t) \in \mathbb{R}^d$ is the Brownian motion (also known as Wiener Process) with dimension $d$, encoding the stochasticity. When the environment is deterministic, we can simply set the $\hat{\Sigma}(s)$ as $\mathbf{0}$. We design the generative model to share the learning control policy with the real environment, as shown in Figure 2. For the inference, the generative model starts from a sample $\hat{s}(0) \in S_0$ and rolls out by drift function $\hat{G}_\alpha$, diffusion function $\hat{\Sigma}_\alpha$, and policy $\pi_\theta$. Therefore, the computation graph contains the learning policy; thus, the auto-differential tools can obtain the gradient for the learning policy by back-propagating through the generative model.

**Remark 1** *We use the fully-connected neural networks to encode such an SDE. Due to the continuity of the neural net, such SDE specification requires the environment dynamics to be continuous and smooth. Therefore our approach cannot handle hybrid dynamics with jump conditions such as the contact dynamics in Mujoco and Safety Gym. Such an assumption is not uncommon, as it remains an open problem to learn the discontinuous dynamics (Parmar et al., 2021; Pfrommer et al., 2021).*

The generative model training is to reduce the following loss function.

$$\min_\alpha \mathcal{L}_g(\tau_\theta, \hat{\tau}_{\theta,\alpha}) = \min_\alpha -\sum_{t=0}^{T} \log\Big( P\Big( s(t) \mid \mathcal{N}(\hat{s}(t), \hat{\Sigma}_\alpha(\hat{s}(t))) \Big) \Big) \tag{3}$$

where $\mathcal{L}_g$ is the maximum likelihood loss, $P\Big( s(t) \mid \mathcal{N}(\hat{s}(t), \hat{\Sigma}_\alpha(\hat{s}(t))) \Big)$ is the likelihood probability of the observed $s(t)$ under the normal distribution of the SDE representation. We use torchsde (Li et al., 2020b) to fit the data $\tau_\theta = \{s(0), s(1), \cdots, s(T)\}$ to the generative model by updating its parameter $\alpha$, which is shown in Lines 2 to 4 in the Algorithm 1.

## 3.3 SOFT BARRIER FUNCTION LEARNING

To encode the hard constraints, we introduce a novel generative-model-based soft barrier function.

**Definition 2** *(Safety Probability Lower Bound) A safety probability lower bound $1 - \eta$ of the entire trajectory (process) $\tau = \{s(0), s(1), \cdots, s(T)\}$ is defined as $P(s(t) \notin S_u | s(0) \in S_0, \forall t \in [0, T]) \geq 1 - \eta$.*

**Definition 3** *(Barrier Function for SDE) Given a policy $\pi_\theta$, $B_\beta$ is a generative-model-based soft barrier function for the discrete-time SDE $\hat{\mathcal{M}}_{\theta,\alpha}$ as in Equation 2, if it is twice differentiable and satisfies the constraints of the lower problem in Equation 1.*

**Lemma 1** *Prajna et al. (2004) Let $B(\hat{s}(t))$ be a supermartingale of the process $\hat{s}(t)$ and $B(\hat{s}) \geq 0, \forall \hat{s} \in S$. Then for any $\hat{s}(0) \in S_0, c > 0$, $P(\sup_{t \geq 0} B(\hat{s}(t)) \geq c \mid \hat{s}(0) \in S_0) \leq \frac{B(\hat{s}(0))}{c}$.*

**Theorem 1** *With a barrier function as in Definition 3, the generative-model SDE with policy $\pi_\theta$ (Equation 2) has a safety probability lower bound $1 - \eta^*$, where $\eta^*$ is the optimal value in the lower problem of Equation 1, as $\forall t \in [0, T], P(\hat{s}(t) \notin S_u | \hat{s}(0) \in S_0) \geq 1 - \eta^*, \hat{s}(t+1) = \hat{\mathcal{M}}_{\theta,\alpha}(\hat{s}(t))$.*

**Proof:** With the last condition of the constraints in the lower problem of Equation 1, we have

$$\mathbb{E}\left[B(\hat{s}(t_2))|\hat{s}(t_1)\right] \leq B(\hat{s}(t_1)), \forall T \geq t_2 \geq t_1 \geq 0,$$

which indicates that the barrier function $B(\hat{s})$ is a supermartingale. Then by leveraging the Lemma 1 above from Prajna et al. (2004), we have

$$P\left(\hat{s}(t) \in S_u, \text{for some } t \in [0, T] \mid \hat{s}(0) \in S_0\right) = P\left(B(\hat{s}(t)) \geq 1, \text{for some } t \in [0, T] \mid \hat{s}(0) \in S_0\right)$$

$$\leq P\left(\sup_{t \in [0,T]} B(\hat{s}(t)) \geq 1 \mid \hat{s}(0) \in S_0\right) \leq B(\hat{s}(0)) \leq \eta^*.$$

Therefore, safety probability lower bound is $1 - \eta^*$ and Theorem 1 holds. $\qquad\square$

We further translate the constraints of the lower problem in Equation 1 with their sampling mean:

$$\min_{\beta} \eta \text{ s.t. } \begin{cases} \frac{1}{N} \sum_{i=1}^{N} B_\beta(\hat{s}^i(0)) \leq \eta, \hat{s}^i(0) \in S_0, & \frac{1}{N} \sum_{i=1}^{N} B_\beta(\hat{s}^i_u) \geq 1, \hat{s}^i_u \in S_u, \\ \frac{1}{N} \sum_{i=1}^{N} B_\beta(\hat{s}^i) \geq 0, \hat{s}^i \in S, \\ \frac{1}{N} \sum_{i=1}^{N} B_\beta(\hat{s}^i(t+1)) \leq B_\beta(\hat{s}^i(t)), \hat{s}^i(t+1) = \hat{\mathcal{M}}_{\theta,\alpha}(\hat{s}^i(t)), \forall t \in [0, T], \forall s^i \in S \setminus S_u. \end{cases}$$

The third non-negative condition is easy to satisfy by setting the output activation function as *Sigmoid* for the barrier neural network. The last condition is to make $B$ as a supermartingale, which is the key to deriving the lower bound of safety probability. In practice, we use a supervised-learning-based method to optimize this problem by minimizing the following loss function:

$$\min_{\theta,\beta} \mathcal{L}_B = \frac{1}{N} \sum_{i=1}^{N} B_\beta(\hat{s}^i(0)) + \frac{1}{N} \sum_{i=1}^{N} (1 - B_\beta(\hat{s}^i_u)) + \frac{1}{N} \sum_{i=1}^{N} \left( \frac{1}{M} \sum_{j=1}^{M} B_\beta(\hat{s}^{i,j}(t+1)) - B_\beta(\hat{s}^i(t)) \right),$$

$$\hat{s}^{i,j}(t+1) = \hat{\mathcal{M}}_{\theta,\alpha}(\hat{s}^i(t)), t \in [0, T], \hat{s}^i(t) \in S \setminus S_u,$$

$$(4)$$

where $\hat{s}^{i,j}(t+1)$ is the next state of $\hat{s}^i(t)$ sampled from the generative model $\hat{\mathcal{M}}_{\theta,\alpha}$ with policy $\pi_\theta$. $\mathcal{L}_B$ essentially reduces the barrier mapping value on $S_0$ (the maximum is $\eta^*(\theta, \alpha)$) and projects the unsafe space $S_u$ to 1 with $Sigmoid$ output, and decreases the expectation of the barrier function along with trajectory. It is worthy to note that $\mathcal{L}_B$ cannot be approximated by the real environment $\mathcal{M}$, as we cannot sample from any intermediate time point $s(t)$ to $s(t+1)$ in the space $S \setminus S_u$ to compute the third sample mean in Equation 4, which is relatively feasible and simple to do with $\hat{\mathcal{M}}_{\theta,\alpha}$ as in Equation 2. The barrier training can be terminated if the second and third sample mean in Equation 4 are non-positive. The soft barrier training is shown as Line 5 in Algorithm 1.

### 3.4 POLICY OPTIMIZATION

As stated before, we use the generative model to generate synthetic data $\hat{\tau}^i_{\theta,\alpha} = \{\hat{s}^i(0), \cdots, \hat{s}^i(T)\} (i \in [1, N])$ with policy $\pi_\theta$ to maximize the total expected return $\hat{J}(\pi_\theta)$ as:

$$\max_{\pi_\theta} \hat{J}(\pi_\theta) = \mathbb{E}_{\hat{s}(0), \hat{\mathcal{M}}_{\theta,\alpha}} \left[ \sum_{t=0}^{T} \gamma^t r\left(\hat{s}(t), \pi_\theta(\hat{s}(t))\right) \right], \text{s.t. } \hat{s}(t+1) = \hat{\mathcal{M}}_{\theta,\alpha}(\hat{s}(t)), \forall t \in [0, T].$$

We use the sample mean from the synthetic trajectories as an estimate for the expectation:

$$\max_{\pi_\theta} \hat{J}(\pi_\theta) = \frac{1}{N} \sum_{i=0}^{N} \sum_{t=0}^{T} \gamma^t r\left(\hat{s}^i(t), \pi_\theta(\hat{s}^i(t))\right), \text{s.t. } \hat{s}^i(t+1) = \hat{\mathcal{M}}_{\theta,\alpha}(\hat{s}^i(t)), \forall t \in [0, T]. \quad (5)$$

With policy $\pi_\theta$ in the forward computation graph of $\hat{\mathcal{M}}_{\theta,\alpha}$, we can directly obtain the backwards gradient for $\pi_\theta$ from Equation 5. The policy optimization is shown as Line 6 in the Algorithm 1.

### 3.5 THEORETICAL ANALYSIS OF SAFETY PROBABILITY BY SOFT BARRIER

For the final learned policy, we conduct a theoretical analysis of its safety probability (as defined in Definition 2), derived from the generative-model-based soft barrier function in our framework.

**Lemma 2** *(Theorem 21 in (Agarwal et al., 2020a))* *Given $\delta \in (0, 1)$, a learned deterministic policy $\pi_\theta(s)$ and assume the environment-policy transition dynamics as $P^*(s'|s) \in \mathcal{P}$ with the function class $|\mathcal{P}| < \infty$ (s' represents the next state of s), let the environment and policy generate a dataset*

*of $n$ trajectories $D := \{(s^j(t), s^j(t+1))\}_{t=0}^T (j = 1, \cdots, n)$, $s(t) \sim D^t = (s^j(0:t-1))$. Note that $D^t$ is a martingale depending on the previous examples. Let the generative model $\hat{\mathcal{M}}_{\theta,\alpha}$ maximize the likelihood of the dataset by its transition dynamics $\hat{P}$ via Equation 3. Then with at least probability $1 - \delta$, the expectation of total variation distance between $P^*$ and $\hat{P}$ is bounded as:*

$$\sum_{t=0}^T \mathbb{E}_{s \sim D^t} \left[ d_{TV}(P^*, \hat{P}) \right] = \sum_{t=0}^T \mathbb{E}_{s \sim D^t} \left\| \hat{P}(s'|s) - P^*(s'|s) \right\|_{\text{TV}}^2 \leq \frac{2 \log(|\mathcal{P}|/\delta)}{n} \qquad (6)$$

**Lemma 3** *(proof provided in the Appendix A) Given a random variable $X_n \geq 0$ on a probability space $\Omega$, if $\mathbb{E}_\Omega[X_n] \to 0$ as $n \to \infty$, then $P(X_n = 0) \to 1$.*

**Proposition 1** *(Asymptotic Lower Bound of Safety Probability) Given the learned policy $\pi_\theta$, let the generative model fit $n$ sample trajectories $\tau_\theta^i (i = 1, \cdots, n)$ from environment $\mathcal{M}$ with $\pi_\theta$ by Equation 3, learn the generative-model-based soft barrier function $B_\beta$ by Equation 4 with $\eta^*$ and assume that it formally satisfies the constraints in Equation 1, then the real environment $\mathcal{M}$ with policy $\pi_\theta$ is safe with at least probability $(1 - \eta^*)$ when $n \to \infty$.*

**Proof of Proposition 1**: Given $(S, \mathcal{B})$ as the measure spaces with $S$ as the state space and $\mathcal{B} = \{B : \mathcal{S} \to \mathbb{R}, \|B\|_\infty \leq 1\}$, where $B$ is a generative-model-based soft barrier function with $Sigmoid$ output, then according to the definition of total variation distance and Lemma 2, we have

$$\sum_{t=0}^T \mathbb{E}_{s \sim D^t} \left[ d_{\text{TV}}(P^*, \hat{P}) \right] = \sum_{t=0}^T \mathbb{E}_{s \sim D^t} \left[ \frac{1}{2} \sup_{B \in \mathcal{B}} \mathbb{E}_{P^*(s'|s)}[B(s')] - \mathbb{E}_{\hat{P}(s'|s)}[B(s')] \right] \leq \frac{2 \log(|\mathcal{P}|/\delta)}{n}.$$

When $n \to \infty$, set $\delta = \frac{1}{n}$, let $X_n = \frac{1}{2} \sup_{B \in \mathcal{B}} \mathbb{E}_{P^*(s'|s)}[B(s')] - \mathbb{E}_{\hat{P}(s'|s)}[B(s')]$, and therefore $\mathbb{E}[X_n] \to 0$. We know $X_n \geq 0$, since $X_n = 0$ when $P^* = \hat{P}$. Therefore, according to Lemma 3, $P(X_n \to 0) \to 1$. We then assume $D^t (t \in [0, T])$ can uniformly cover the space $S$ as $n \to \infty$, thus the soft barrier becomes a true barrier function for the real environment and Proposition 1 holds. $\square$

**Remark 2** *(Practical Safety Probability Lower Bound) In addition to the asymptotic safety probability, we propose a finite-sample practical safety probability lower bound. We first sample the generative model and the environment with the final learned policy to quantify their maximum distance per state as $\Delta = \max_{t \in [0,T], i=1, \cdots, N} |s^i(t) - \hat{s}^i(t)|$, and then enlarge the unsafe region with $\Delta$ by Minkowski sum as $S'_u = S_u \bigoplus \Delta$. Next, we retrain another generative-model-based soft barrier function $B$ with $S'_u$. Finally, we conservatively report $(1 - \max_{(\hat{s} \in \hat{\tau}_t^i, i=1, \cdots, N)} B(\hat{s}_t^i))$ as the final lower bound of safety probability by the soft barrier function.*

**Remark 3** *(During-learning Safety) The above asymptotic and practical safety bounds are derived for the final learned policy. It is possible that $1 - \eta^*$ is not a valid safety probability bound during learning, as there exist a modeling gap between the generative model and the real environment. However, we optimize $1 - \eta^*$ during learning to increase the chance of finding safer learned policies at the end, as demonstrated in our experiments below.*

## 4 EXPERIMENTAL RESULTS

**Experiment Settings and Examples:** We compare our approach with two state-of-the-art open-source CMDP-based methods, PPO-L (Ray et al., 2019) and FOCOPS (Zhang et al., 2020). For these two baselines, we design the cost function such that the state is safe if its cost is less than 0. It is worth noting that PPO-L has a stronger safety constraint than FOCOPS as we implemented the PPO-L with the expectation of cost per state as $\mathbb{E}[c(s, a)] \leq 0$, rather than the cumulative cost in FOCOPS as $\mathbb{E}\left[\sum_{t=0}^T c(s, a) \leq D'\right]$. In FOCOPS, We conservatively set $D' = -60$ for the 2D and cartpole examples below, and $-200$ for the Rocket and UAV examples, to improve its safety. We mark this safety-oriented version FOCOPS*. We mainly compare the converged final policy of each method in system safe rate measured via simulations – we call it *empirical safe rate*. We

Table 1: Comparison of our approach with CMDP-based baselines PPO-L and FOCOPS*. $s_e$ is the safe rate by simulating 500 random initial states from $S_0$. $1 - \eta$ is the practical lower bound of safety probability in our approach as $(1 - \max_{(\hat{s} \in \hat{\tau}_t^i, i=1,\cdots,n)} B(\hat{s}_t^i))$, derived by **Remark 2**. Our approach achieves significantly higher $s_e$ than the baselines. It is observed that $1 - \eta$ is a lower bound of $s_e$.

| Metric | Methods | 2D | Cartpole | Rocket | UAV |
|---|---|---|---|---|---|
| $s_e$, empirical safe rate | Ours | **99.9(0.09)%** | **100%** | **100%** | 99.6(0.3)% |
| | PPO-L | 98.9(0.08)% | 89.3(5.5)% | 96.4(6.3)% | **100%** |
| | FOCOPS* | 98.7(0.18)% | 84.2(4)% | **100%** | 91(4.2)% |
| $1 - \eta$, lower bound | Ours | 97.6(1.3)% | 86.6(2.9)% | 89.9(1.6)% | 93.2(2.2)% |
| | PPO-L, FOCOPS* | - | - | - | - |
| $J(\pi)$, performance | Ours | -67.3(4.9) | -24.4(4.7) | **-143.2**(1.6) | -847.1(6.5) |
| | PPO-L | **-66.3**(5.3) | -34.1(6.7) | -151.4(3.6) | -895.5(4.3) |
| | FOCOPS* | -69.8(3.2) | **-15.2**(2.3) | -249.1(1.4) | **-734.1**(3.3) |

also perform safety probability analysis for our method (CMDP cannot provide one), and compare different methods in total reward return.

Note that learning safe control policy for high-dimensional systems is quite challenging. Current state-of-the-art works of certificate-based policy learning mainly focus on low-dimensional systems with fewer than 6 dimensional states (Luo & Ma, 2021; Lindemann et al., 2021; Chang et al., 2019; Berkenkamp et al., 2017). In this paper, we test our approach on 13D UAV and Rocket examples.

*2-Dimensional SDE (Prajna et al., 2004)* has the unknown dynamics $\mathcal{M}$ as $\dot{s_1} = 0.8 s_2$, $\mathrm{d}s_2 = (a - 0.3 s_1^3)dt + 0.2\mathrm{d}W(t)$ ($W(t)$, Wiener process.) Initial space $S_0 = \{(s_1 + 2)^2 + s_2 \leq 0.01\}$, and unsafe space $S_u = \{s_1 \in [-1, 0], s_2 \in [1.2, 1.7]\}$. The goal is to stabilize the system near $(0, 0)$.

*Cartpole Balancing (Brockman et al., 2016)* has a 4-dimensional vector $s = [x, \theta, \dot{x}, \dot{\theta}]$ as the system state, where $x$ is the position and $\theta$ is the angular error to the upright. The initial space $S_0 = \{(x, \theta, \dot{x}, \dot{\theta}) | x \in [-0.167, 0.033], \theta \in [-0.6, -0.5], \dot{x} = -0.35, \dot{\theta} = 0.53\}$, and unsafe space $S_u = \{(x, \theta, \dot{x}, \dot{\theta}) \mid x \leq -0.75\}$. The goal is to keep the cartpole balanced upright.

*Powered Rocket Landing (Jin et al., 2021)* has 6 DoF (degrees of freedom) with 13 system states and 3 action variables. The goal is to land the rocket close to the original point while avoiding an unsafe region. Its state vector is $\mathbf{s} = [\mathbf{p} \ \mathbf{v} \ \mathbf{q} \ \omega] \in \mathbb{R}^{13}$, where $\mathbf{p} = (x, y, z) \in \mathbb{R}^3$ and $\mathbf{v} = (v_x, v_y, v_z) \in \mathbb{R}^3$ represent the position and velocity of the rocket, respectively. $\mathbf{q} \in \mathbb{R}^4$ is the unit quaternion for attitude and $\omega \in \mathbb{R}^3$ is the angular velocity with respect to the inertial frame. There are three trust forces for the rocket as the control input $\mathbf{u} = [T_x, T_y, T_z] \in \mathbb{R}^3$. The initial space $S_0 : \mathbf{p} = (x, y, z)(x - 10)^2 + (y + 8)^2 + (z - 5)^2 \leq 0.01, \mathbf{v} = 0, \mathbf{q} = (0.73, 0, 0, 0.68), \omega = 0$, and unsafe space $S_u : \mathbf{p} = (x, y, z)(x - 5)^2 + y^2 \leq 1, -2 \leq z \leq 5, \|\mathbf{v}\|_1 \leq 10, \|\omega\|_1 \leq 10$.

*UAV Maneuvering (Jin et al., 2021)* is to maneuver an UAV close to the original point while avoiding an obstacle. The 6-DoF UAV has 13 system states and 4 action variables. Its state vector is $\mathbf{s} = [\mathbf{p} \ \mathbf{v} \ \mathbf{q} \ \omega] \in \mathbb{R}^{13}$, same with above Rocket example. The control input $\mathbf{u} = [T_1, T_2, T_3, T_4] \in \mathbb{R}^4$ includes the four rotating propellers of the quadrotor. $S_0 : \mathbf{p} = (x, y, z)(x + 8)^2 + (y + 6)^2 + (z - 9)^2 \leq 0.01, \mathbf{v} = 0, \mathbf{q} = (1, 0, 0, 0), \omega = 0, S_u : \mathbf{p} = (x, y, z)(x + 4.5)^2 + (y + 4)^2 \leq 1, -2 \leq z \leq 5$.

**Comparison and Effectiveness of Our Approach:** Table 1 shows the comparison results in simulation-based system safe rate (based on 500 simulations for each example, with random initial states), safety probability, and performance. We can see that by directly enforcing hard safety constraints via soft barrier functions, **our approach can achieve significantly higher system safe rate than the CMDP-based baselines**. Our approach also provides a practical lower bound of safety probability, which the CMDP-based methods cannot provide. CMDP achieves better performance (total reward return) in some cases, but we view safety as the first priority for these systems and the focus of this work.

Figure 3 shows the control trajectories by the learned policies from our approach and the baselines. The agent is always safe with our learned policy, while there exist unsafe cases by both PPO-L and FOCOPS. Moreover, our generative model behaves very similarly to the real environment, which shows the usefulness of the generative modeling for constructing the soft barrier function and opti-

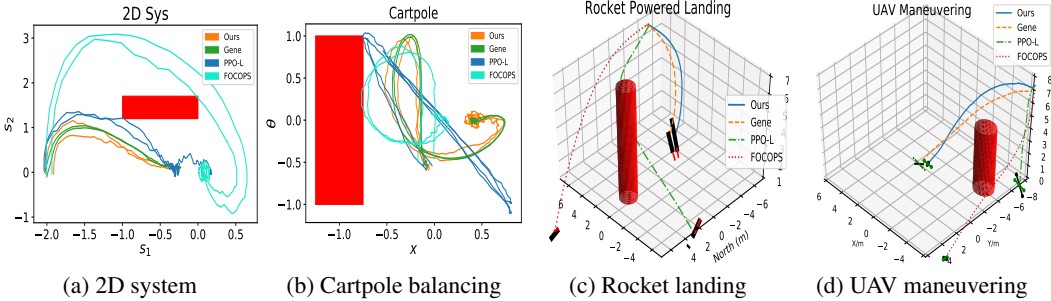

(a) 2D system      (b) Cartpole balancing      (c) Rocket landing      (d) UAV maneuvering

Figure 3: Control trajectories by the learned policies from our approaches and baselines. "Gene" indicates the synthetic trajectory from the final learned generative model, which behaves very similarly to the real environment with "Ours" policy, showing its effectiveness for barrier function construction. We can see that our approach learns safer policies than the baselines.

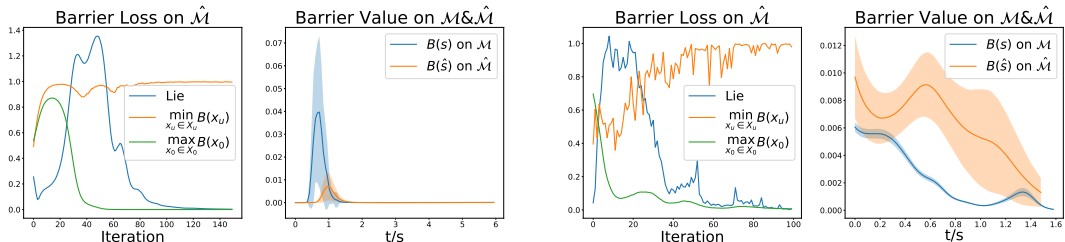

Figure 4: Barrier function training and testing in the 2D SDE example.

Figure 5: Barrier function training and testing in the UAV example.

mizing the control policy. We also show the learning process of the soft barrier function on the generative model and its testing in the real environment in Figures 4 and 5 for the 2D system and UAV examples (results of the other two examples are in the Appendix A). The learned barrier function maps the initial space to near $0$ and the unsafe space to $1$ with the third sample mean in Equation 4 to $0$ (marked as Lie in Figures). The barrier function has a similar value along with the trajectories in the real environment and the generative model. Again, this indicates that the generative model behaves very similarly to the environment, as shown in Figure 3. Although the barrier function decreases or stays constant most of the time, it can increase at some point. This is due to that 1) the possible modeling error between the generative model SDE and environment 2) the supervised learning cannot cover all possible cases for barrier training.

**Limitations:** As stated earlier, one key assumption of this work is the smoothness and continuity of the system behavior, which prevents its application to hybrid dynamics with jump conditions such as the contact dynamics in Mojuco and Safety Gym. One possible solution is to learn an ensemble generative model as a hybrid system to deal with those discontinuous contact dynamics, and we plan to explore it in future work. Another limitation of our framework is on the computation complexity of the generative model (e.g., it takes around 8 hours to learn a policy for the Cartpole example and 1 day for the UAV and Rocket examples). In future work, we plan to improve the efficiency of this part by exploring techniques such as Continuous Latent Process Flows (CLPF) Deng et al. (2021).

## 5 CONCLUSION

We present a safe RL approach for unknown stochastic environment that enforces hard reachability-based safety constraints through generative-model-based soft barrier functions. Our approach formulates a novel bi-level optimization formulation, and develops a safe RL algorithm that jointly learns the generative model, soft barrier function, and policy optimization. Experiments demonstrate that our approach can significantly improve empirical system safe rate over CMDP-based baselines and also provide a practical lower bound of safety probability.

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

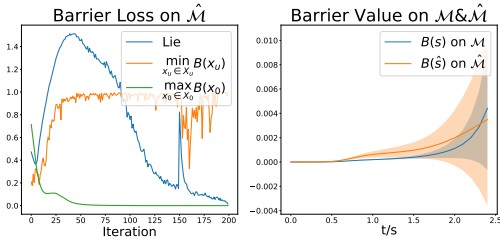

Figure 6: Barrier function training and testing in Rocket powered landing.

Figure 7: Barrier function training and testing in Cartpole balancing.

# A APPENDIX

## A.1 PROOF OF LEMMA 3 IN THE MAIN TEXT

For any $m \in \mathbb{N}$, let $E_m = \{\omega \in \Omega : X_n(w) > \frac{1}{m}\}$. Since $X_n \geq 0$, we have:

$$\mathbb{E}_\Omega[X_n] = \int_\Omega X_n \mathrm{d}P \geq \int_{E_m} X_n \mathrm{d}P \geq \frac{1}{m} P(E_m).$$

Therefore, $P(E_m) \to 0$, and then:

$$0 \leq P(\omega \in \Omega : X_n(w) \neq 0) = P(\bigcup E_m) = \lim_{m \to \infty} P(E_m) \to 0,$$
$$P(\omega \in \Omega : X_n(w) \neq 0) \to 0 \implies P(\omega \in \Omega : X_n(w) = 0) \to 1.$$

## A.2 ADDITIONAL EXPERIMENTAL RESULTS

The barrier function training and testing results for the Rocket powered landing and the Cartpole balancing examples are shown here in Figures 6 and 7.

