# OpenReview forum: "Enforcing Hard Constraints with Soft Barriers: Safe Reinforcement Learning in Unknown Stochastic Environments"
_ICLR.cc/2023/Conference — Submitted to ICLR 2023_

### Official Review · Reviewer_H8ou · 2022-10-18

**Confidence:** 4
**Correctness:** 3
**Technical Novelty And Significance:** 3
**Empirical Novelty And Significance:** 2
**Recommendation:** 6

**Clarity, Quality, Novelty And Reproducibility:**

### Clarity
This paper is overall organized and easy to follow. I feel Section 3.1 is a bit messy and relatively hard to follow, but I don't have an idea for improving it.

### Quality
- As I mentioned in Weakness, the quality of related work section should be improved by surveying and referring more papers (especially on safe model-based RL). Also, in experiments section, the authors' proposed method should be compared with more baselines (e.g., Yarden et al. (2021) in more complicated environments (e.g., Safety-Gym).

### Novelty
- As far as I know, this proposed method is novel, but generative model as a form of world models has already exists; hence, several sentences should be modified.

### Reproducibility
- The source code is not attached and it is not sure whether it is open-sourced upon publications, so I need to say that reproducibility is low for now.


**Strength And Weaknesses:**

### Strength
- Important and relatively-unexplored problem settings on safe RL with **hard constraints**.
- New algorithms based on shielding and generative models
- Good experimental results compared to CMDP methods and FOCOPS*.
- Overall, this paper is well-organized and easy to follow.

### Weakness
- Safe model-based RL
    - There are significantly few references to safe model-based RL. I know there is the following sentence, but still I consider the relevant methods should be mentioned in related work section and compared to the authors' proposed method.
    - > "However, these methods do not apply to our setting, where the environment is uncertain and stochastic without explicit dynamics, backup safe policy, or human intervention."
    - Hard constraints in safe RL has been studied in unknown (stochastic) environment. For example, Moldovan and Abbeel (2012) studied safe exploration in unknown environment in terms of ergodicity. Also, Turchetta et al. (2016) and Wachi et al. (2018) studied safe exploration problems in unknown environment while guaranteeing reachability and returnability under regularity assumptions. I think the authors should have surveyed wider safe RL methods (the three papers I mentioned are all hard safety constraint; that is, the agent is required to satisfy the safety constraint at every time step).
    - Though the above papers are indeed different from the authors' one in terms of problem formulation or assumptions, Yarden et al. (2021) can be mentioned in related work section and compared to the authors' method in the experiments. If I understand correctly, the method in Yarden et al. (2021) can be fairly compared to the authors' one, and source code is also open-sourced (https://github.com/yardenas/la-mbda),

- Experiments
    - Benchmark problems are very easy. Hopefully, Safety-Gym should be used for testing safe **deep** RL methods.
    - Computational time and sample efficiency should be also compared. I "guess" the authors' proposed method requires much more data until convergence.

### Minor comments and typos
- [Section 3.1. Equation regarding $J$] I think $t$ should be $i$ (or $i$ should be $t$).
- [Definition 2] $\tau(t)$ is not defined. Is $\tau(t) := [ s(i) ]_{i=t}^T$?

### References
- Moldovan, T., and Abbeel, P. 2012. Safe exploration in Markov decision processes. In International Conference on Machine Learning (ICML).
- Turchetta, M.; Berkenkamp, F.; and Krause, A. 2016. Safe
exploration in finite Markov decision processes with gaussian processes. In Advances In Neural Information Processing Systems, 4305–4313.
- Wachi, Akifumi, et al. "Safe exploration and optimization of constrained mdps using gaussian processes." Proceedings of the AAAI Conference on Artificial Intelligence. Vol. 32. No. 1. 2018.
- As, Yarden, et al. "Constrained Policy Optimization via Bayesian World Models." International Conference on Learning Representations. 2021.


**Summary Of The Paper:**

This paper is on safe reinforcement learning (RL) with unknown stochastic environments. Under the safety definition in terms of reachability, the authors aim to enforce hard constraints rather than soft constraints that are often the main target of constrained Markov decision processes (CMDP) literature. In this paper, the authors introduce a notion of generative model based soft barrier functions to encode the hard safety constraints. The authors propose a safe RL algorithm to jointly model the environment and optimize the agent policy while satisfying the safety (reachability) constraints. Finally, this paper demonstrate the effectiveness of the proposed method in several benchmark problems.

**Summary Of The Review:**

I think this paper has several merits in terms of 1) important problem settings, 2) new algorithm, 3) good experimental results (easy benchmark and limited baselines), and 4) good writing.

However, I have several concerns in terms of 1) how to deal with existing safe model-based RL literature and 2) empirical experiments, as stated in Weakness.
Therefore, I recommend rejection.

---

> ### Author Response · Authors · 2022-11-19
> **To Reviewer H8ou 1/2**
>
> _Q1. There are significantly few references to safe model-based RL. I know there is the following sentence, but still I consider the relevant methods should be mentioned in the related work section and compared to the authors' proposed method._
>
> * Thanks for the comment. The model-based safe RL methods, particularly those leveraging control theoretic approaches such as the control barrier functions, typically require information on one or more of the following aspects: 1) explicit known dynamics models, 2) a safe initial guess of control policy, and 3) inputs from human experts. These assumptions lead to different problem settings as our work, and those algorithms cannot be easily applied to our problem. In the revision, we have added some of those model-based safe RL algorithms in the related works (“Model-based Safe RL by Formal Methods”).
>
> _Q2. Hard constraints in safe RL has been studied in the unknown (stochastic) environment. For example, Moldovan and Abbeel (2012) studied safe exploration in unknown environment in terms of ergodicity. Also, Turchetta et al. (2016) and Wachi et al. (2018) studied safe exploration problems in unknown environment while guaranteeing reachability and returnability under regularity assumptions. I think the authors should have surveyed wider safe RL methods (the three papers I mentioned are all hard safety constraint; that is, the agent is required to satisfy the safety constraint at every time step)._
>
> * In the revision, we have added [Moldoval et al. 2012], [Turchetta et al. 2016] and [Wachi et al. 2018] in the Introduction section. We explained that while these works address hard safety constraints, they focus on discrete state and action spaces where the hard safety constraints are defined as a set of unsafe state-action pairs that cannot be visited. This is different from our problem, where we address continuous control for stochastic and continuous unknown MDPs.
>
> _Q3. Though the above papers are indeed different from the authors' one in terms of problem formulation or assumptions, Yarden et al. (2021) can be mentioned in related work section and compared to the authors' method in the experiments. If I understand correctly, the method in Yarden et al. (2021) can be fairly compared to the authors' one, and source code is also open-sourced (https://github.com/yardenas/la-mbda)_
>
> * The paper [Yarden et al. 2021] uses image observation as the state input and learns latent dynamics through a World model [Danijar et al. 2019]. Due to the image input, we cannot find a way to directly compare it with our work since it is unclear how to learn the transition dynamics between images for our approach. While this work is a model-based approach, the safety constraints are formulated in the CMDP manner, therefore we have added it in the revision under “Safe RL by CMDP” in the Related Work section.

---

> > ### Comment · Reviewer_H8ou · 2022-11-20
> > **Thanks for the comments**
> >
> > I appreciate the authors' response.
> >
> > The authors address most of the concerns I had at the timing of initial review.
> > I think the paper gets substantially improved; thus, I increase the score from 5 to 6.

---

> > > ### Author Response · Authors · 2022-11-21
> > > **Thanks for the response**
> > >
> > > Thank you very much for checking our response and raising the score. We appreciate your comments which are greatly helpful for our paper's improvement.

---

> ### Author Response · Authors · 2022-11-19
> **To Reviewer H8ou 2/2**
>
> _Q4. Benchmark problems are very easy. Hopefully, Safety-Gym should be used for testing safe deep RL methods. Computational time and sample efficiency should be also compared. I "guess" the authors' proposed method requires much more data until convergence._
>
> * Learning safe control policy for high-dimensional systems is quite challenging. Current state-of-the-art certificate-based policy learning mainly focuses on low-dimensional systems with fewer than 6-dimensional states (Luo & Ma, 2021; Lindemann et al., 2021; Chang et al., 2019; Berkenkamp et al., 2017). In this paper, we successfully test our approach on 13-dimensional UAV and Rocket examples, which is much more complex than the previous works. However, one key assumption of this work is the smoothness and continuity of the system behavior, which prevents its application to hybrid dynamics with jump conditions such as the contact dynamics in Mojuco and Safety Gym. One possible solution is to learn an ensemble generative model as a hybrid system to deal with those discontinuous contact dynamics, and we plan to explore it in future work. In the revision, we have added a discussion of benchmark dimensionality and the limitation of our current approach in requiring continuous and smooth dynamics (Section 4).
>
> * In terms of sample efficiency, as a model-based approach, we train the barrier function and safe policy jointly based on synthetic data from the generative model, and we only train the generative model with sample data from the environment. For instance, in the Cartpole example, the amount of sample data from the environment only reaches 6.4e5 (based on batch size 64, control steps 100, and N = 100 in Algorithm 1), which is less than the 1e6 needed for the CMDP approaches tested. If we count the synthetic data from the generative model, our approach will indeed consume much more data than the CMDP approaches, but accessing such synthetic data is typically much “cheaper” than interacting with the real environment in practice. In terms of computational time, the generative model training in the inner loop of Algorithm 1 for Cartpole (i from 0 to 100) is around 250s, while the barrier training and policy optimization take around 50s. Overall it takes around 8 hrs to obtain the final learned policy for Cartpole. The current computational bottleneck is at the forwarding and backwarding process of the generative model (we use torchSDE). We have added a brief discussion of computational time in our revision (Section 4, Limitations) and plan to improve it in our future work.
>
> _Q5.  I feel Section 3.1 is a bit messy and relatively hard to follow. The source code is not attached and it is not sure whether it is open-sourced upon publications, so I need to say that reproducibility is low for now._
>
> * We have made changes to notations, symbols, definitions, and proofs to improve paper readability, updated related works and experimental results, and improved paper writing in general. We are currently cleaning our code repo and post it online as soon as possible.

---

### Official Review · Reviewer_MNvg · 2022-10-21

**Confidence:** 3
**Correctness:** 3
**Technical Novelty And Significance:** 3
**Empirical Novelty And Significance:** 3
**Recommendation:** 8

**Clarity, Quality, Novelty And Reproducibility:**

As mentioned above, the clarity is a substantial issue with this paper. For example, it seems that the notation changes at various points in the proof of the one theoretical claim of the paper (that or the objects under discussion appear without a proper introduction).  Along these lines, I am really not sure what l(tau,\hat{\tau}) means in the generative model loss function -- a likelihood is obtained for a realization against a distribution (model), but this seems to be a likelihood between two realizations. I am guessing that \hat{\tau} represents the distribution of the model, but that's just a wild guess.

Likewise, I can't make any sense of the plots in Figures 4-5---what's on the x-axis? And, what method is used to obtain the "gene" trajectory in Figure 3?

Otherwise, the quality and novelty of the work seem fairly strong, as noted above.

**Strength And Weaknesses:**

The main strengths of the paper are (1) the approach to learning the barrier function seems to be fairly novel in this setting and (2) the experiments do seem to be pretty promising. In particular, on the technical side, I did appreciate the use of the (locally-defined) supermartingale constraint on the barrier function to obtain the safety bound.

The main weakness is that the paper is a bit of a mess -- I'll discuss this below.

An asymptotic theoretical guarantee is included, which is better than nothing, but certainly much weaker than we'd want to have if the objective was to ensure safety. Remark 1 proposes a "practical safety probability lower bound" that is used to obtain bounds reported in the experiments, that are indeed less than the empirical fraction of safe trajectories (1), but it isn't immediately obvious that the bound is valid.

**Summary Of The Paper:**

The focus of this work is on learning safe MDP policies, in the sense that a safety constraint holds across the entire trajectory of the policy. The distinction from the widely-studied CMDPs is that the constraint in the CMDP holds over a cost aggregated across the entire trajectory, and not at each point in the trajectory. The approach in this work is to learn a barrier function that is a supermartingale over the filtration defined by transitions, and thus with high probability is not violated during the execution. There is a (weak) theorem showing that in the asymptotic limit, the policy achieves safety. More compellingly, experiments on a few control benchmarks suggest that the method indeed avoids unsafe regions.

**Summary Of The Review:**

The paper's presentation is rough, but the approach has some significant technical novelty and seems to be effective empirically.

---

> ### Author Response · Authors · 2022-11-19
> **To Reviewer MNvg**
>
> _Q1. An asymptotic theoretical guarantee is included, which is better than nothing, but certainly much weaker than we'd want to have if the objective was to ensure safety. Remark 1 proposes a "practical safety probability lower bound" that is used to obtain bounds reported in the experiments, that are indeed less than the empirical fraction of safe trajectories (1), but it isn't immediately obvious that the bound is valid._
>
> * Thanks for the comment. It remains an open and hard problem to quantify the modeling gap between the generative model and the environment under limited sample data, and it is not clear to us how to derive non-asymptotic safety bound under such uncertain modeling error. And because of such a gap, our method indeed cannot ensure that the safety bound is valid during learning, and we have clarified this in the revision (Remark 3). However, for the final learned policy and dynamics, the $1-\gamma$ safety probability should well reflect the real trajectory safety probability under a large amount of sampled data – even with a large time horizon, as the barrier function of SDE ensures the safety probability by sup operator within the horizon. In our experiments (Table 1), we have also shown that the generative model behaves very similarly to the real environment in Figures 3-7 (updated in the revision) and the barrier function based on the generative model applies well to the real trajectories from the environment. Those results demonstrate that the practical safety lower bound is valid with a high probability for the final learned policy.
>
> _Q2. As mentioned above, the clarity is a substantial issue with this paper. For example, it seems that the notation changes at various points in the proof of the one theoretical claim of the paper (that or the objects under discussion appear without a proper introduction). Along these lines, I am really not sure what l(tau,\hat{\tau}) means in the generative model loss function -- a likelihood is obtained for a realization against a distribution (model), but this seems to be a likelihood between two realizations. I am guessing that \hat{\tau} represents the distribution of the model, but that's just a wild guess. Likewise, I can't make any sense of the plots in Figures 4-5---what's on the x-axis? And, what method is used to obtain the "gene" trajectory in Figure 3?_
>
> * We apologize for the writing quality of the original submission. In the revision, we have updated notations, symbols, definitions, and proofs to improve paper readability. We have added an explanation of the maximum likelihood loss function, where at time $t$, $s(t)$ is the observed realization data under the normal distribution of the generative model $\mathcal{N}(\hat{s}, \hat{\Sigma}(\hat{s}(t)))$ (based on the SDE). We have also updated the figures. The x-axis of the left subplot in Figures 4 and 5 denotes the training iteration of the barrier function given the policy and generative model. The x-axis of the right subplot in Figures 4 and 5 denotes the control time of the examples in seconds. “Gene” in Figure 3 indicates the synthetic trajectory from the final learned generative model, which behaves very similarly to the real environment with “Ours” policy, showing its effectiveness for barrier function construction.

---

> ### Author Response · Authors · 2022-11-21
> **A Gentle Reminder for Further Discussion to Reviewer MNvg**
>
> Thank you again for your reviews. We hope our response can address your comments and questions related to the safety bound and paper writing.  We take this as a great opportunity to improve our work and appreciate any additional feedback from you.

---

> ### Author Response · Authors · 2022-12-05
> **Looking forward to further discussion and feedback - reviewer MNvg**
>
> We deeply appreciate your valuable time and efforts spent reviewing this paper and providing constructive comments in the initial review. It would be very much appreciated if you could once again help review our responses and the updated paper which addresses the writing problems. Please let us know if the response and updated paper writing address or partially address your concerns and if our explanations are heading in the right direction. Please also let us know if there are further questions or comments about this paper. We strive to improve the paper consistently, and it is our pleasure to have your feedback!

---

### Official Review · Reviewer_YtE9 · 2022-10-25

**Confidence:** 3
**Correctness:** 3
**Technical Novelty And Significance:** 2
**Empirical Novelty And Significance:** 3
**Recommendation:** 5

**Clarity, Quality, Novelty And Reproducibility:**

Currently the paper is quite hard to follow. There are lots of notations in various parts of the proofs that have not been formally defined and well explained. Also some notations seem to be repetitive and not well defined (e.g., \tau), and for example the generative model loss is not clearly defined.  This makes checking the theoretical proofs of this work rather difficult. Several descriptions at the plots are also unclear or missing, which makes parsing the results difficult.

While some experimental setup has been discussed, it does not seem sufficient for readers to reproduce the results without the source code pf these methods. Thus this affects reproducibility.

**Strength And Weaknesses:**

Strengths:
The barrier-based method for guaranteeing safety  is novel and is related to the work of guaranteeing safety in the form of stability in model-based RL algorithms.
The problem of enforcing RL policy to be safe w.r.t. to hard constraints is interesting and important (in areas e.g., robotics).
The algorithms proposed is justified with theoretical studies (which utilize martingale properties of the barrier function to obtain a safety results in the asymptotic sense) and its benefits are also illustrated with some experiments over standard CMDP methods.

Weaknesses:
It seems the safety bound only holds in asymptotic, and it is unclear whether this method ensures the property of safety during training.
The experiments seem to be overly simple. Running this algorithm on more realistic benchmarks (e.g., Safety gym for safe RL experimentations) and doing more comparisons with other model-based safe RL methods will be better in terms of demonstrating the effectiveness of this method.
It is unclear whether the safety bounds provided in the theory can be justified by the experimental results
The quality on the writing of the proofs is currently quite low, it requires more work to explain the notations and intuitions clearly in order to make it readable.

**Summary Of The Paper:**

In this paper, the authors proposed ways to do safe RL in an unknown stochastic environment. Specifically, they model the problem in hard constraints instead of the standard CMDP constraints and derived a novel barrier function method to incorporate the hard constraints into the RL algorithm. The authors manage to show with theoretical results that such an approach asymptotically achieves safety in policy optimization in RL, and they demonstrate the effectiveness of the proposed method on several safe RL benchmark problems for which the proposed method outperform SOTA in terms of safety guarantees and return performance.


**Summary Of The Review:**

This paper studied an important safe RL problem and proposed a novel barrier-function based method. It also showed this method's effectiveness both theoretically (in asymptotic sense) as well as empirically on several safety RL benchmarks. However, it'd be great to verify the benefits of this method by 1) testing in more complex domains and 2) comparing with more model-based safe RL methods (e.g., Turchetta, M. et al Safe exploration in finite Markov decision processes with gaussian processes. NeurIPS 16). The writing of this paper also needs significant improvements for it to be readable/understandable by the broader ML audience. The current paper is quite hard to follow.

---

> ### Author Response · Authors · 2022-11-19
> **To Reviewer YtE9**
>
> _Q1. Weaknesses: It seems the safety bound only holds in asymptotic, and it is unclear whether this method ensures the property of safety during training. The experiments seem to be overly simple. Running this algorithm on more realistic benchmarks (e.g., Safety gym for safe RL experimentations) and doing more comparisons with other model-based safe RL methods will be better in terms of demonstrating the effectiveness of this method. It is unclear whether the safety bounds provided in the theory can be justified by the experimental results. The quality on the writing of the proofs is currently quite low, it requires more work to explain the notations and intuitions clearly in order to make it readable._
>
> * Thanks for the comment. The safety bound in the paper is asymptotic because the generative model is asymptotically getting close to the real environment. However, it remains an open and hard problem to quantify the modeling gap between the generative model and the environment under limited sample data. Because of such a gap, our method indeed cannot ensure that the safety bound is valid during learning. We have clarified this in the revision (Remark 3). However, for the final learned policy and dynamics, the $1-\gamma$ safety probability should well reflect the real trajectory safety probability under a large amount of sampled data – even with a large time horizon, as the barrier function of SDE ensures the safety probability by sup operator within the horizon. In our experiments (Table 1), we have also shown that the generative model behaves very similarly to the real environment in Figures 3-7 (updated in the revision) and the barrier function based on the generative model applies well to the real trajectories from the environment.
>
> * In terms of benchmarks, learning safe control policy for high-dimensional systems is quite challenging. Current state-of-the-art certificate-based policy learning mainly focuses on low-dimensional systems with fewer than 6-dimensional states (Luo & Ma, 2021; Lindemann et al., 2021; Chang et al., 2019; Berkenkamp et al., 2017). In this paper, we successfully test our approach on 13-dimensional UAV and Rocket examples, which is much more complex than the previous works. However, one key assumption of this work is the smoothness and continuity of the system behavior, which prevents its application to hybrid dynamics with jump conditions such as the contact dynamics in Mojuco and Safety Gym. One possible solution is to learn an ensemble generative model as a hybrid system to deal with those discontinuous contact dynamics, and we plan to explore it in future work. In the revision, we have added a discussion of benchmark dimensionality and the limitation of our current approach in requiring continuous and smooth dynamics (Section 4).
>
> * To the best of our knowledge, there is no model-based safe RL approach in the reference that can be applied to our problem setting where the environment is a general continuous stochastic unknown MDP.
>
> * Sorry about the writing quality. In the revision, we have made changes to notations, symbols, definitions, and proofs to improve paper readability.
>
> _Q2. Clarity, Quality, Novelty And Reproducibility:
> Currently the paper is quite hard to follow... Thus this affects reproducibility._
>
> * In the revision, we have updated notations, symbols, definitions, proofs, and descriptions of the plots to improve paper readability, including the definition of $\tau$ and generative model loss. We will post our code repo online as soon as possible.
>
> _Q3. However, it'd be great to verify the benefits of this method by 1) testing in more complex domains and 2) comparing with more model-based safe RL methods (e.g., Turchetta, M. et al Safe exploration in finite Markov decision processes with gaussian processes. NeurIPS 16). The writing of this paper also needs significant improvements for it to be readable/understandable by the broader ML audience._
>
> * The UAV and Rocket examples are much more complex than previous formal-certificate-based RL approaches, and we have added a short discussion in the experiments section in the revision.
>
> * The model-based safe RL methods, particularly those leveraging control theoretic approaches such as the control barrier functions, typically require information on one or more of the following aspects: 1) explicit known dynamics models, 2) a safe initial guess of control policy, and 3) inputs from human experts. These assumptions lead to different problem settings as our work, and those algorithms cannot be easily applied to our problem. For instance, the reference [Turchetta et al. 2016] assumes known deterministic dynamics for safe RL, which cannot be applied to our stochastic environment setting. In the revision, we have added some of those model-based safe RL algorithms in the related works (“Model-based Safe RL by Formal Methods”).

---

> ### Author Response · Authors · 2022-11-21
> **A gentle reminder for further discussion to reviewer YtE9**
>
> Thank you again for your effort in providing the reviews. We hope our response below is able to address your comments and questions on during-learning safety, experiments, and paper writing. Please let us know if you have further suggestions and feedback. We appreciate this opportunity to further improve our work.

---

> ### Author Response · Authors · 2022-12-05
> **Looking forward to the discussion and further feedback - reviewer YtE9**
>
> The authors sincerely look forward to your feedback as the discussion period is closing soon. We deeply appreciate your valuable time and efforts spent reviewing this paper and the constructive comments in the initial review. It would be much appreciated if you could once again help review our responses and let us know if these address or partially address your concerns and if our explanations are heading in the right direction. Please also let us know if there are further questions or comments about the updated paper. We strive to improve the paper consistently, and it is our pleasure to have your feedback!

---

### Official Review · Reviewer_kxrF · 2022-11-04

**Confidence:** 4
**Correctness:** 2
**Technical Novelty And Significance:** 3
**Empirical Novelty And Significance:** Not applicable
**Recommendation:** 5

**Clarity, Quality, Novelty And Reproducibility:**

The studied problem of safe RL with bounded constraints satisfaction is an interesting topic and has gained significant attention in the learning and control community. The paper is in general well written to highlight the motivation, existing challenges, and the adapted approach of the safe RL framework with soft barrier function learning. With that being said, however, the technical details seem insufficient to justify the theoretical guarantee of the probabilistic safety and it is unclear how the algorithms would compare with those existing approaches that are not based on CMDP framework. Please see detailed comments as follows.

- Contribution and Novelty: the contribution of the proposed soft barrier functions and the bi-level optimization based safe RL framework is overall well presented, and the authors have made a good point in comparison to other CMDP based approaches. However, as mentioned it is unclear how the algorithms would compare to model-based learning with control theoretic approaches for safety considerations. Although it is claimed in the related work that “these approaches either require explicit known system models for barrier or shielding construction, or an initial safe policy…”, the following work did not require either of them:
[1] Berkenkamp, F., Turchetta, M., Schoellig, A., & Krause, A. (2017). Safe model-based reinforcement learning with stability guarantees. Advances in neural information processing systems, 30.
	[2] Taylor, Andrew, Andrew Singletary, Yisong Yue, and Aaron Ames. "Learning for safety-critical control with control barrier functions." In Learning for Dynamics and Control, pp. 708-717. PMLR, 2020.

- Quality: one major concern is the technical quality in terms of proof of safety for the proposed bi-level optimization process. Given the unknown and stochastic environment, it is challenging to learn both barrier functions and the optimal policy safely without making further assumptions, e.g. bounded Lipschitz continuity, etc. With the presented analysis, it is unclear to the reviewer that (a) if the barrier functions are always learnable before violating a safety constraint, and (b) if the probability of the entire trajectory is always bounded with a satisfying probability due to unbounded stochastic noise. In the carpole example, a valid safe control policy might not exist if the cartpole is already at the boundary of a safety constraint with some initial velocity. And when the time horizon is very large, it seems intractable to bound the safety probability given the cascading effect of the step-wise unsafe probability. More discussions on this point could be helpful to make the safety conclusion more convincing.

- The experimental results are reported with few trials and no quantitative performance in terms of mean and variances are given to justify the robustness of the performance.


**Strength And Weaknesses:**

Strength:
+ The general idea of this paper is well written with clear motivation and good overview about the high level idea of the algorithm.
+ The bi-level optimization framework that jointly learns the policy and soft barrier functions is interesting.

Weakness:
- While the reviewer found the comparative analysis with CMDP algorithm is sound and solid, it is unclear how the algorithm could compare with other model-based safe RL algorithms, especially methods that combine RL and control theoretic approaches such as control barrier functions.
- The related work discussion needs to be improved, as some claims seem inaccurate given existing approaches that do not require explicit known model nor an initial safe policy, e.g. [Berkenkamp et al 2017].
- The proof of safety (Lemma 1) is substandard in absence of supplementary materials. For example, it is unclear how the conclusion of bounded probability for the ENTIRE trajectory could be reached with the presented proof, especially given the stochastic environment.
- The experimental results are reported with few trials and no quantitative performance in terms of mean and variances are given to justify the robustness of the performance.


**Summary Of The Paper:**

This paper addresses the problem of safe reinforcement learning with hard constraints in an unknown stochastic environment. The key idea (and the contribution) is the proposed generative model based soft barrier functions that relax the hard safety constraints adapting to the unknown environment. This is further incorporated into the presented safe RL framework as a bi-level optimization process for jointly learning the optimal control policy, the soft barrier functions, and the initially unknown environment model, with avoidance of unsafe region in a probabilistic setting. Experimental results with two baseline approaches such as PPO-L and FOCOPS are presented to show the effectiveness of the proposed algorithm.

**Summary Of The Review:**

The paper presents an interesting topic of safe RL in an unknown and stochastic environment. While the general idea is clear, the reviewer feels that the technical part should be substantially improved given the insufficient details regarding safety guarantee, which is the core part of the presented approach. The experimental results also seems substandard without extensive trials under different environment settings. Authors are encouraged to address the comments in the future version of the paper.

---

> ### Author Response · Authors · 2022-11-19
> **To Reviewer kxrF 1/2**
>
> _Q1. While the reviewer found the comparative analysis with CMDP algorithm is sound and solid, it is unclear how the algorithm could compare with other model-based safe RL algorithms, especially methods that combine RL and control theoretic approaches such as control barrier functions._
> * Thanks for the comment. The model-based safe RL algorithms, particularly those leveraging control theoretic approaches such as the control barrier functions, typically require information on one or more of the following aspects: 1) explicit known dynamics models [Choi et al 2020], 2) a safe initial guess of control policy [Berkenkamp et al 2017], and 3) inputs from human experts [Xu et al 2022]. These assumptions lead to different problem settings as our work, and those algorithms cannot be easily applied to our problem. In the revision, we have added some of those model-based safe RL algorithms in the related works (“Model-based Safe RL by Formal Methods”).
>
> 1. Choi, Jason, et al. "Reinforcement Learning for Safety-Critical Control under Model Uncertainty, using Control Lyapunov Functions and Control Barrier Functions." RSS, 2020.
> 2. Felix Berkenkamp, Matteo Turchetta, Angela Schoellig, and Andreas Krause. “Safe Model-based Reinforcement Learning with Stability Guarantees.” Neurips, 2017.
> 3. Xu, Yunkun, et al. "Look Before You Leap: Safe Model-Based Reinforcement Learning with Human Intervention." CoRL, 2021.
>
> _Q2. The related work discussion needs to be improved, as some claims seem inaccurate given existing approaches that do not require explicit known model nor an initial safe policy, e.g. [Berkenkamp et al 2017]._
>
> * The reference [Berkenkamp et al. 2017] requires an initial safe policy, as its Introduction states “In particular, we show how, starting from an initial, safe policy we can expand our estimate of the region of attraction by collecting data inside the safe region and adapt the policy to both increase the region of attraction and improve control performance.’’ The requirement of an initial safe policy is also highlighted in its Background and Assumptions sections. Moreover, the approach also requires a nominal dynamical model as shown in Eq.(1), where the h function is known.
>
> _Q3. The proof of safety (Lemma 1) is substandard in absence of supplementary materials. For example, it is unclear how the conclusion of bounded probability for the ENTIRE trajectory could be reached with the presented proof, especially given the stochastic environment._
>
> * In the revision, we have updated this proof and also reorganized the paper to make it more clear – the original Lemma 2 has been moved ahead as the new Lemma 1, and the original proof of safety in Lemma 1 has been renamed as Theorem 1 since it provides a standalone conclusion for the generative model.
> In the proof, we first show that $B(\hat{s})$ is a supermartingale based on the definition of barrier function (condition 4 in the lower problem of Eq(1)). Then, with the updated Lemma 1 from reference [Prajna et al. 2004], we show that
> $$
> P(\hat{s}(t) \in S_u, \textrm{for some t} \in [0, T] | \hat{s}(0) \in S_0) = P(B(\hat{s}(t) \geq 1, \textrm{for some t} \in [0, T] | \hat{s}(0) \in S_0))
>  \leq P(\sup_{t \in [0, T]}B(\hat{s}(t)) \geq 1 | \hat{s}(0) \in S_0)) \leq B(\hat{s}(0)) \leq \eta^*.
> $$
> Here $\hat{s}(t)$ is the state from the generative model as an SDE, and the safety bound of the ENTIRE trajectory by the generative model is achieved by the $sup_{t \in [0, T]}$ operator. This safety bound of the entire trajectory is for the generative model as the barrier function is built on it.
>
> _Q4. The experimental results are reported with few trials and no quantitative performance in terms of mean and variances are given to justify the robustness of the performance._
> * In the revision, we have updated Table 1 with the mean and std values reported from 5 individual runs.

---

> ### Author Response · Authors · 2022-11-19
> **To Reviewer kxrF 2/2**
>
> _Q5. Contribution and Novelty: the contribution of the proposed soft barrier functions and the bi-level optimization based safe RL framework is overall well presented, and the authors have made a good point in comparison to other CMDP based approaches. However, as mentioned it is unclear how the algorithms would compare to model-based learning with control theoretic approaches for safety considerations. Although it is claimed in the related work that “these approaches either require explicit known system models for barrier or shielding construction, or an initial safe policy…”, the following work did not require either of them: [1] Berkenkamp, F., Turchetta, M., Schoellig, A., & Krause, A. (2017). Safe model-based reinforcement learning with stability guarantees. Advances in neural information processing systems, 30. [2] Taylor, Andrew, Andrew Singletary, Yisong Yue, and Aaron Ames. "Learning for safety-critical control with control barrier functions." In Learning for Dynamics and Control, pp. 708-717. PMLR, 2020._
>
> * Thanks for the comment. As stated above, the reference [Berkenkamp et al. 2017] requires an initial safe policy, as its Introduction states “In particular, we show how, starting from an initial, safe policy we can expand our estimate of the region of attraction by collecting data inside the safe region and adapt the policy to both increase the region of attraction and improve control performance.’’ The requirement of an initial safe policy is also highlighted in its Background and Assumptions sections. Moreover, the approach also requires a nominal dynamical model as shown in Eq.(1), where the h function is known. The reference [Taylor et al. 2020] requires a nominal model in Eq.(4), which further supports Eq.(6). Moreover, the system dynamics considered there is deterministic (and also a special case that can be expressed in a control-affine form), which differs from our general stochastic MDPs. In the revision, we have added these two works in the related works, along with several others on model-based safe RL algorithms.
>
> _Q6. Quality: one major concern is the technical quality in terms of proof of safety for the proposed bi-level optimization process. Given the unknown and stochastic environment, it is challenging to learn both barrier functions and the optimal policy safely without making further assumptions, e.g. bounded Lipschitz continuity, etc. With the presented analysis, it is unclear to the reviewer that (a) if the barrier functions are always learnable before violating a safety constraint, and (b) if the probability of the entire trajectory is always bounded with a satisfying probability due to unbounded stochastic noise. In the carpole example, a valid safe control policy might not exist if the cartpole is already at the boundary of a safety constraint with some initial velocity. And when the time horizon is very large, it seems intractable to bound the safety probability given the cascading effect of the step-wise unsafe probability._
>
> * Thanks for the comment. In the revision, we have updated Section 3 to clarify the assumptions and limitations of our approach.
>
> * First, in Remark 1 and Section 3.1, we explain that the underlying dynamics of the environment is assumed as continuous and smooth without jump dynamics (contact dynamics). This is because we abstract the generative model as an SDE by neural network representation, whose continuity requires the environment not to contain any discontinuity. This assumption is common in learning-based dynamics modeling as it remains a challenging problem for a neural network to learn a discontinuous function, as stated in Remark 1. Then, as we consider general continuous stochastic unknown environments, (a) we cannot ensure that the barrier functions are always learnable before violating a safety constraint,  and (b) we cannot ensure that the probability of the entire trajectory is always bounded with a satisfying probability due to unbounded stochastic noise (the noise in our paper can be unbounded as it is expected to be a Wiener process, which is a Gaussian at every timestep).
>
> * In Remark 3, we explained that the safety probability presented in the paper is for the final learned policy and barrier function. The system may violate the safety probability bound during the learning process due to the modeling gap between the generative model and the real environment under limited interaction data. However, for the final learned policy and dynamics, the $1-\gamma$ safety probability should well reflect the real trajectory safety probability under a large amount of sampled data – even with a large time horizon, as the barrier function of SDE ensures the safety probability by sup operator within the horizon.
>
> * Finally, for the provided cart pole example, if a valid safe control does not exist, the safety probability $1-\gamma$ would be 0 as it is impossible to separate the safe and unsafe space by the barrier function.

---

> ### Author Response · Authors · 2022-11-21
> **A gentle reminder for further discussion to reviewer kxrF**
>
> Thank you again for your initial comments. Please let us know whether our responses have addressed your questions and comments in the initial reviews. We shall be grateful if you have further suggestions and feedback. We take it as an opportunity to further improve this paper.

---

> ### Author Response · Authors · 2022-12-05
> **Gentle reminder to reviewer kxrF**
>
> As the discussion period is closing soon, we sincerely look forward to your further feedback. The authors deeply appreciate your valuable time and efforts spent reviewing this paper and helping us improve it. It would be very much appreciated if you could once again help review our responses and let us know if these address or partially address your concerns and if our explanations are heading in the right direction. Please also let us know if there are further questions or comments about this paper. We strive to improve the paper consistently, and it is our pleasure to have your feedback!

---

### Author Response · Authors · 2022-11-18
**To all the reviewers**

We would like to thank all the reviewers for their insightful comments and constructive suggestions. We have uploaded a revised version of our submission, with major changes highlighted in blue (there are also other minor changes in wording and typos). According to the review feedback, we have made changes to notations, symbols, definitions, and proofs to improve paper readability, updated related works and experimental results, and improved paper writing in general. Below we provide our detailed response to each reviewer.

---

### Decision · Program_Chairs · 2023-01-20

**Decision:**

Reject

**Justification For Why Not Higher Score:**

Lack of clarity in writing which makes it difficult to properly evaluate the results. Simple experiments that do not support the theoretical findings appropriately. The theoretical results are only asymptotic which is weak if the objective is to ensure safety.

**Justification For Why Not Lower Score:**

N/A

**Metareview: Summary, Strengths And Weaknesses:**

The reviewers believe the paper studies an important problem, the proposed method is novel, and the technical side of the paper is strong. However, there are concerns about

1) The clarity of the paper. The reviewers see it as a major issue. The paper is not easy to follow and it is hard to properly verify the results. There are undefined notations and notation changes in the proofs. There is not enough information about the experimental set up to allow a reader to reproduce the results.
2) The derived safety bound is only asymptotic, which is weak when the goal is to ensure safety. If the bound is only valid when the generative model gets (asymptotically) close to the real environment, then the title of the paper "safe RL in unknown stochastic environments" seems to be slightly misleading.
3) The problems used in the experiments are overly simple and the results do not properly support the theoretical findings.

I believe the paper has potential but it is not ready for publication and needs to be improved. I would strongly recommend that the authors take the reviewers' comments into consideration, revise their work, improve its quality, and prepare it for future conferences.